# Cumulative human impacts on global marine fauna highlight risk to biological and functional diversity

**Casey C. O'Hara**[1]*, **Melanie Frazier**[1], **Mireia Valle**[1,2,3], **Nathalie Butt**[4,5],
**Kristin Kaschner**[6], **Carissa Klein**[5], **Benjamin S. Halpern**[1,7]

**1** National Center for Ecological Analysis and Synthesis, University of California, Santa Barbara, Santa Barbara, California, United States of America, **2** AZTI, Marine Research, Basque Research and Technology Alliance (BRTA), Sukarrieta, Spain, **3** Basque Centre for Climate Change (BC3), Scientific Campus of the University of the Basque Country (UPV-EHU), Leioa, Spain, **4** The Nature Conservancy, South Brisbane, Queensland, Australia, **5** Centre for Biodiversity and Conservation Science, School of the Environment, The University of Queensland, Brisbane, Queensland, Australia, **6** Department of Biometry and Environmental System Analysis, Albert-Ludwigs-University of Freiburg, Freiburg, Germany, **7** Bren School of Environmental Science & Management, University of California, Santa Barbara, Santa Barbara, California, United States of America

* ohara@nceas.ucsb.edu

**Data Availability Statement:** All data used as inputs for this assessment are freely available from original sources. All original code and data resulting from this analysis has been deposited at

## Abstract

Anthropogenic stressors to marine ecosystems from climate change and human activities increase extinction risk of species, disrupt ecosystem integrity, and threaten important ecosystem services. Addressing these stressors requires understanding where and to what extent they are impacting marine biological and functional diversity. We model cumulative risk of human impact upon 21,159 marine animal species by combining information on species-level vulnerability and spatial exposure to a range of anthropogenic stressors. We apply this species-level assessment of human impacts to examine patterns of species-stressor interactions within taxonomic groups. We then spatially map impacts across the global ocean, identifying locations where climate-driven impacts overlap with fishing, shipping, and land-based stressors to help inform conservation needs and opportunities. Comparing species-level modeled impacts to those based on marine habitats that represent important marine ecosystems, we find that even relatively untouched habitats may still be home to species at elevated risk, and that many species-rich coastal regions may be at greater risk than indicated from habitat-based methods alone. Finally, we incorporate a trait-based metric of functional diversity to identify where impacts to functionally unique species might pose greater risk to community structure and ecosystem integrity. These complementary lenses of species, function, and habitat provide a richer understanding of threats to marine biodiversity to help inform efforts to meet conservation targets and ensure sustainability of nature's contributions to people.

Figshare. DOI: 10.6084/m9.figshare.26454106. A living version of the code and data can be found at https://github.com/mapping-marine-spp-vuln/spp_vuln_mapping.

**Funding:** CK, BSH: National Philanthropic Trust (https://www.nptrust.org/), SB200099 CCO: National Science Foundation via Belmont Forum (https://belmontforum.org/about), 2019902 No funders played any role in study design, data collection, analysis, decision to publish, or preparation of the manuscript.

**Competing interests:** The authors have declared that no competing interests exist.

## Introduction

Anthropogenic stressors from human activities on land and sea coupled with those from anthropogenic climate change are driving degradation of coastal ecosystems, marine regime shifts, and increased extinction risk of threatened species [1–6], threatening the biological and functional diversity that underpin ecosystem services upon which we depend [7–11]. The Kunming-Montreal Global Biodiversity Framework [12] calls for urgent action to reduce extinction risk, improve ecosystem integrity, and ensure sustainability of nature's contributions to people. To these ends, understanding where and to what degree anthropogenic stressors impose impacts on marine biological and functional diversity will be critical to designing, allocating, and monitoring effective conservation actions at scale [13].

To assess the risk of adverse impacts on marine ecosystems, cumulative human impact models have been developed to integrate information on vulnerability of elements of ecological concern (e.g., habitats, species, functional groups) and exposure of those elements to anthropogenic stressors [14]. Habitat-based estimates of ecosystem vulnerability to (e.g., [15]) and impact from (e.g., [7, 14, 16]) various stressors rely upon an understanding of fundamental structural and functional similarity between, say, a Caribbean coral reef and an Indonesian coral reef, or a Californian kelp forest and a Namibian kelp forest ecosystem, though there may be little overlap in the species that inhabit each system. Such a habitat-based approach provides valuable insights on potential impacts to the general trophic structure and functioning of an ecosystem as well as the types of ecosystem services that may be affected, but it may miss important differences in vulnerability stemming from heterogeneity of species composition between otherwise similar marine communities.

A species-based approach to estimating impacts may be better able to capture heterogeneity of species' vulnerability to stressors [3, 17–19], providing insights on impacts to individual species and community structure. Studies assessing anthropogenic impacts on marine species (e.g., [18, 20–23]) generally have been limited in scope to a few select taxa and/or stressors, largely due to lack of a systematic means of estimating species vulnerability across multiple taxa and stressors [19, 24]. O'Hara et al [3] parsed threat information from IUCN Red List assessments to estimate sensitivity and impacts from multiple anthropogenic stressors across multiple marine taxa globally over an eleven-year period, but that study was necessarily restricted to those species categorized as threatened and near-threatened, excluding the vast majority of marine species. However, a recently developed trait-based framework to estimate species vulnerability broadly across taxa and stressors [19] greatly expands the set of marine species available for such cumulative impact assessment. Additionally, a species-based approach to estimating risk of impact, coupled with information on traits associated with ecological function, enables identification of functionally unique species and groups whose loss may pose greater risk to ecosystem functioning and resilience than similar impacts on functionally redundant species [25–28].

Here we provide a taxonomically-diverse spatial analysis of cumulative human impacts of 13 anthropogenic stressors on 21,159 marine animal species and subpopulations. For each species/stressor combination, we intersect the species' range with the spatial distribution of the stressor; impact in each pixel is modeled as the product of stressor intensity and the species' estimated vulnerability to that stressor (See S1 Fig in S1 File. for conceptual overview of methods). We spatially aggregate species impact distributions to estimate mean impact across species and taxa, providing a taxonomically detailed understanding of how anthropogenic pressures impose risk of impact to marine biodiversity. We then compare these results to cumulative impact estimates based on representative habitats to determine where and how species- vs. ecosystem-level vulnerability drives potential impact. Finally, we use a set of traits

to classify species into functional entities (*sensu* Mouillot et al. [26]) and use these groupings to calculate a weighted mean impact, emphasizing impacts to species who uniquely represent a position in functional trait space. The species, functional entity, and habitat approaches to estimating cumulative impact provide different but synergistic lenses through which to estimate our impact on marine ecosystems: areas of agreement between these methods reinforce urgency for conservation, while areas of significant difference may provide conservation insights by highlighting impacts on vulnerable and functionally important species in otherwise resilient marine ecosystems.

## Methods

### Analysis grid

All spatial analyses were calculated on a gridded global map using a Mollweide equal-area projection coordinate reference system (CRS), gridded to 10 km x 10 km resolution. See SI Methods in S1 File for additional details on preparing the analysis grid.

### Species distributions

The 21,159 species (including subpopulations) considered in this assessment are limited to those animal species with data on spatial distribution as well as sufficient trait data to estimate vulnerability and assign species to functional entities. These species represent only a small subset of the >240,000 marine species identified in the World Register of Marine Species (WoRMS, [29]); however, this subset includes most known marine mammals, marine reptiles, seabirds, and cartilaginous fishes, as well as about half of marine bony fishes and warm-water corals (S1 Table in S1 File by class, S2 Table in S1 File vertebrates by order). Together these species represent most top predators, many mid-trophic species, and ecologically critical habitat-forming species. Relatively fewer other invertebrates were included, as most lacked spatial data, trait data, or both.

Species distribution data were taken from AquaMaps [30] (n = 18,480) and IUCN species distribution maps [31, 32] (n = 2,679). For species appearing in both distribution map datasets, the AquaMaps distribution maps, based on transparent and repeatable algorithms using publicly available data, were preferred over IUCN range maps, which integrate data and expert knowledge but may include mapping decisions that are difficult to replicate. For species represented by the AquaMaps dataset, presence was calculated as any 0.5˚ cell with a probability of occurrence of 0.5 or greater; the resulting cells were then reprojected to the 10 km Mollweide analysis grid. For species represented by IUCN Red List rangemaps, the polygons were reprojected and rasterized to the resolution and CRS of the analysis grid. See SI Methods in S1 File for additional details on preparing species distributions. See S2 Fig in S1 File. for a map of species richness generated from these species distributions.

### Vulnerability estimates

Vulnerability weights, i.e., the relative effect of a given stressor on the fitness/health of a given species, were determined based on methods of Butt et al. [19]. Briefly, that study estimated vulnerability of species to each of a suite of stressors based on presence of certain traits that are likely to increase the species' physiological sensitivity (e.g., calcium external structures indicate higher sensitivity to ocean acidification), ability to adapt to or avoid that specific stressor (e.g., high mobility makes it easier to avoid localized stressors), and life history and population-level traits that affect the population's ability to adapt to or recover from disturbances in general (e.g., high fecundity suggests easier recovery from a disturbance). A binary exposure multiplier

(zero or one) prevents nonsensical results for certain stressors where exposure is limited to certain depths or ocean zones, e.g., ship strikes will not affect mesopelagic species. Trait values for species were gathered through expert elicitation and provided as ordinal or nominal categorical values. Vulnerability weights range from 0 (a stressor does not affect a species) to 1 (a stressor imposes extreme adverse effects on a species).

See SI Methods in S1 File and S3 Table in S1 File for details on traits and calculations.

### Stressor layers for species-focused analysis

For the species-focused analysis, the intensity of exposure to a stressor depends on the spatial distribution of the stressor relative to the spatial distribution of the species. Spatial data for stressors is typically available as gridded data of some physical quantity related to anthropogenic activity, e.g., brightness of nighttime lights, tonnes of nutrient fertilizer runoff, population density within 25 km of coast, or value of aragonite saturation state. For each stressor, a reference value was determined from the data (typically 99.9th percentile of observed values), a historic baseline (e.g., mean/standard deviation of sea surface temperature from 1985–2015), or ecologically relevant value (e.g., aragonite saturation state of 1) (S4 Table in S1 File). We calculated stressor distributions as a value from 0 (stressor not present) to 1 (stressor at reference point, indicating maximum intensity).

For most of the included stressors, a single map of relative stressor intensity was created from gridded data and applied to all species, although vulnerability to the stressor varied by species. These stressors include sea surface temperature (SST) extremes, ocean acidification, ultraviolet radiation, sea level rise, nutrient pollution (runoff), direct human disturbance, light pollution, shipping (ship strikes), and habitat destruction driven by demersal destructive fishing and the footprint of benthic structures.

However, there were also several stressors for which intensity (again ranging from 0 to 1) depends on species-specific information. These stressors include bycatch (dependent on water column position, i.e., benthic, pelagic, or both), biomass removal (dependent on catch that is directly targeting that species), and increase in mean SST (dependent on species thermal tolerance). For these stressors, individual maps were generated for each species (biomass removal, SST rise) or for each water-column position category (bycatch).

See SI Methods in S1 File and S4 Table in S1 File for details on the data source, transformation, and reference point used for these stressor layers.

### Stressor layers for habitat-focused analysis

The habitat-focused analysis was similar to that for species, with the intensity of exposure to a stressor depending on the spatial distribution of the stressor relative to the distribution of the habitat. For this approach, fisheries stressors were calculated using the same source as the species-level stressors, i.e., Watson [33], but aggregated by fishing gear, depth, and scale according to their effects on various habitat types as described in Halpern et al. [7]: commercial pelagic and demersal low bycatch, commercial pelagic high bycatch, commercial demersal destructive, and artisanal/small scale fishing. For SST extremes, ocean acidification, ultraviolet radiation, sea level rise, nutrient pollution (runoff), direct human disturbance, light pollution, shipping, benthic structures, and demersal destructive fishing, we used the identical stressor layers prepared for the species-level analysis. The species-specific stress of increasing mean SST relative to their thermal tolerance was omitted, as it would not be feasible to determine an analogous habitat-level thermal tolerance. See SI Methods in S1 File and S4 Table in S1 File for details on the data source, transformation, and reference point used for these stressor layers.

## Cumulative human impacts: Species method

**Estimating impact at species level per grid cell.** We modeled the impact on species $i$ of stressor $j \in 1:J$ in a given location (i.e., grid cell) as the product of stressor intensity $s_j$ and vulnerability of that species to that stressor $v_{ij}$:

$$I_i^j = v_{ij}s_j$$

Cumulative impact on species $i$ in a given location was determined by summing impacts across all stressors (or subset, e.g., climate vs. non-climate stressors) in that location:

$$I_i^{cml} = \sum_{j=1}^{J} v_{ij}s_j$$

Note that this additive model does not account for compound effects of multiple stressors acting in combination, i.e., synergistic or antagonistic effects. Meta-analyses examining two-stressor interactions [34, 35] have observed some non-additive stressor interactions, but additive effects were more commonly reported. Additionally, an additive model requires fewer assumptions, is conceptually tractable, and likely results in more conservative results.

**Estimating species-level mean cumulative impact across species range.** For each species $i$, we calculated a cumulative impact score $X$ accounting for impacts across its entire range as an average of per-grid-cell impacts for all cells $c$ in the species' range $R_i$. For a single stressor $j$:

$$X_i^j = \frac{1}{R_i} \sum_{c \in R_i} I_{ic}^j$$

Cumulative impact scores across multiple stressors (climate, non-climate, and total) were determined as the sum of single-stressor impact scores.

**Estimating impact across species per grid cell.** The species-mean method for calculating the impact score for stressor $j$ in a given cell was determined by taking an unweighted mean across all $N$ species present (or a taxonomic subset, e.g., all elasmobranchs):

$$I_{spp}^j = \frac{1}{N} \sum_{i=1}^{N} v_{ij}s_j$$

and the cumulative impact is the sum of impacts across all (or a subset of) stressors within that cell.

## Cumulative human impacts: Habitat method

To compare the results of our species-based cumulative impact approach to those of a habitat-based approach (e.g., [14]; [7]), we recreated habitat maps at the same resolution and projection as the species-based analysis, aggregating habitat presence maps at ~1 km resolution to determine proportional habitat representation within each 10 km grid cell. Using these habitat maps, we applied habitat vulnerability weights from Halpern et al. [7] to determine impacts based on largely the same stressor maps data sources used for the species-based assessment.

To identify vulnerability of each habitat to various stressors we used the matrix of habitat vulnerability from Halpern et al. [7].

Per-grid-cell habitat impact scores for each stressor $j$ were created as the product of habitat vulnerability for each habitat $h \in 1: H$ and intensity of stressor $j$, averaged over the proportional

inclusion of that habitat $p_h$ in a given cell:

$$I_{hab}^j = \sum_{h=1}^{H} p_h v_{hj}$$

Cumulative impact per pixel is the sum of habitat-based impacts across all (or subset) of stressors.

## Cumulative human impacts: Functional entity method

**Functional entities.** To estimate cumulative impact on functional diversity, we first assigned species to functional entities based on categorical values of four traits (maximum body length, adult mobility, position in water column, and adult trophic level) that roughly determine a species' ecological niche with regard to regulation of food webs and nutrient cycling, following Mouillot et al. [26]. Due to limited trait data available across a broad range of taxa, we relied on a smaller set of traits (those four noted previously) for assignment of functional entity than the six traits used in Mouillot et al. [26], resulting in fewer but more populous functional entities and therefore a more conservative estimate of functional vulnerability.

Trait values were gleaned from [19, 36, 37]; missing values were imputed using Multiple Imputation by Chained Equation (MICE) in the R package mice [38] using all other traits plus fecundity (where available), generation time (where available), order, and family. See SI Methods in S1 File for details on the trait values used to assign functional entities, along with analyses to test sensitivity of functional vulnerability and cumulative impact to potential error in imputation of traits.

**Estimating impact at functional entity level per grid cell.** For each functional entity $k \in 1$: $K$ consisting of some subset of species in a particular location, the impact of stressor $j$ on the functional entity is the mean impact across all species in that functional entity in that location:

$$I_k^j = \frac{1}{N_{FE}} \sum_{i=1}^{N_{FE}} I_i^j$$

Cumulative impact of all stressors on this functional entity in this location is the sum of impacts across all stressors (or a subset).

**Estimating impact across functional entities per grid cell.** The functional entity method for calculating the impact score for stressor $j$ in a given location was determined by taking a weighted mean across all $K$ functional entities present. Weighting for each functional entity was based on the functional vulnerability, *sensu* Mouillot et al. [26] with a slight modification (see below).

$$I_{FE}^j = \frac{1}{\sum_{k=1}^{K} FV_k} \sum_{k=1}^{K} FV_k I_k^j$$

Mouillot et al. [26] scored vulnerability of a functional entity as 1 if that entity was represented by a single species and 0 otherwise. Here we calculated functional vulnerability based on an inverse exponential of the number of species that represent that functional entity in that location, where functional vulnerability of entity $k$ was calculated as $FV_k = \left(\frac{1}{2}\right)^{N_k - 1}$, accounting for low-membership entities but rapidly asymptotically approaching zero as membership increases.

As for the species-based approach, the cumulative impact is the sum of impacts across all (or a subset of) stressors within that cell.

## Code and packages

All analysis was performed in R statistical software, version 4.0.4 [39], relying primarily on packages tidyverse [40], terra [41], sf [42], taxize [43, 44], rfishbase [45].

## Results

Mean impacts on the assessed species varied dramatically across and within taxa (Fig 1A–1C). Comparing across taxa, mean risk of impact was highest for corals, followed by other invertebrate groups, driven in large part by higher vulnerability to increasing sea surface temperature and ocean acidification. Of vertebrate taxa, elasmobranchs were on average at greatest risk, driven by rising temperatures and high fishing pressure. For species and taxa whose range extended beyond the continental shelf, coastal impacts from non-climate stressors were generally higher than when assessed across their full range (i.e., suggesting relatively lower impacts away from the continental shelf), though coastal and full-range climate stressors were of similar magnitude. Average impact scores for vertebrate taxa fell below those of invertebrates, though there is considerable variation within each taxon, resulting in outliers: the top 1% of

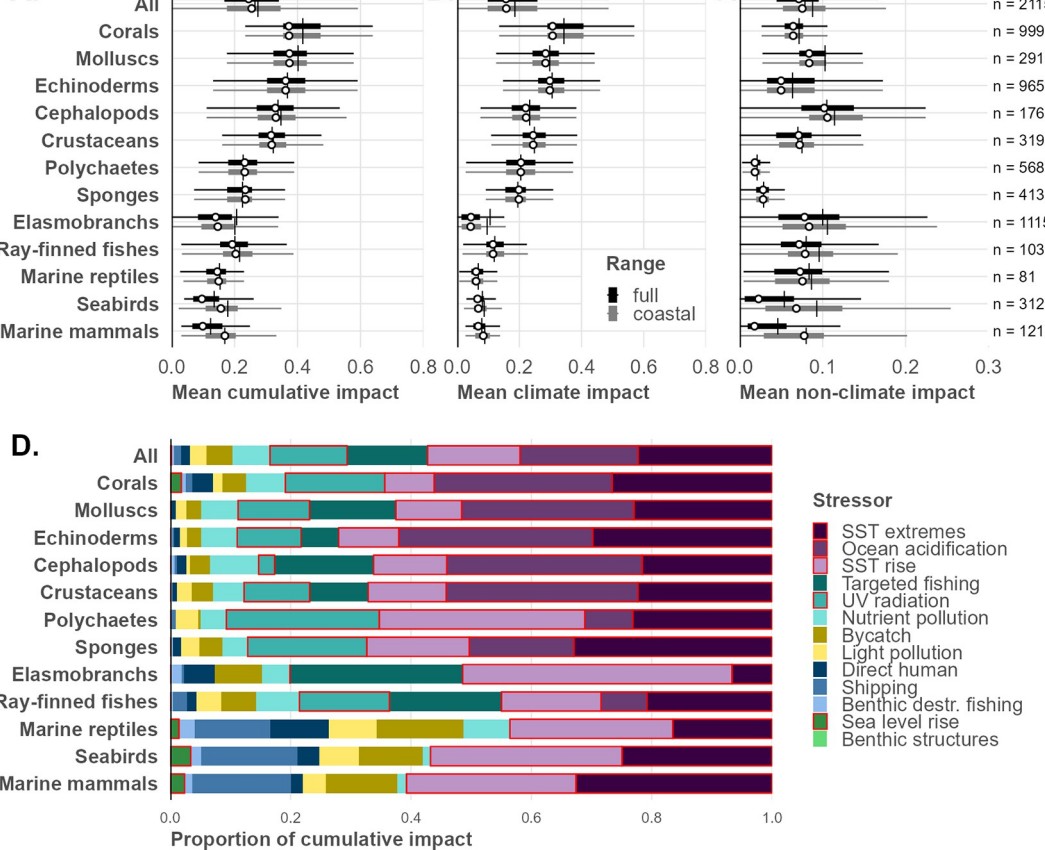

**Fig 1. Cumulative productivity-weighted mean impacts on species ranges by taxon.** (A) Distribution of mean predicted impact across species ranges (full range and coastal portions of range) from anthropogenic stressors by taxonomic group. (B) Distribution of mean cumulative impact from five climate stressors. (C) Distribution of mean cumulative impact from eight non-climate stressors. Vertical black line indicates mean across all species in taxon; white point indicates median. Boxes represent interquartile range (IQR, quartile Q1 to Q3); whiskers indicate observations 1.5x IQR below (above) Q1 (Q3) of box; outliers omitted from plot for clarity. (D) Contribution of individual stressors to mean cumulative impact across species ranges by taxon. Climate stressors outlined in red.

species by impact score (n = 211 of 21,159) contain a disproportionately high number of elasmobranchs (n = 66, 31.3% of the highest impact subset), molluscs (n = 70, 33.2%), and corals (n = 13, 6.1%) relative to their representation in the overall dataset (5.3%, 13.8%, and 4.7% respectively). The bottom 1% of species by impact score also contain a disproportionate number of elasmobranchs (n = 46, 21.8% of the lowest impact subset), as well as polychaetes (n = 22, 10.4%) and ray-finned fishes (n = 140, 66.4%).

Sea surface temperature rise (long-term trends) and extremes (short-term events, e.g., marine heat waves) were substantial contributors to impacts across all taxa (Fig 1D). Ocean acidification was a dominant stressor on all invertebrate species, but imposes little to no direct effect on vertebrates. Ultraviolet radiation impacted most invertebrates primarily due to vulnerability of planktonic larval life stages. Targeted fishing imposed the greatest risk of impact of all non-climate stressors on average, even considering that many taxa are not targeted and therefore not directly impacted (and thus are scored as zero in the calculation of mean impact). Fisheries bycatch impacted species broadly across all taxa.

Spatial distribution of impacts is calculated as the mean impact from all stressors, or some subset of stressors, across all species present in each analysis cell. Cumulative impacts tend to be highest along coastlines, particularly heavily populated coastal zones where human populations generate more localized stressors while overlapping with higher magnitude of diffuse climate stressors (e.g., Northern European waters, the Mediterranean Sea, and the Indo-Pacific region, Fig 2A). Lower cumulative impacts were observed in remote areas where low fishing and shipping activity coincided with relatively low climate stressors (e.g., Southeastern Australia, equatorial Eastern Pacific, Southern Atlantic, Northern Atlantic near Greenland). The very lowest cumulative impacts were observed in areas dominated by permanent or seasonal sea ice (Arctic Ocean, coastal Antarctica). Climate stressors broadly impact species across coastal and open ocean regions, with highest values in tropical Indo-Pacific waters, the Caribbean Sea, and northern Pacific and Atlantic (Fig 2B). Non-climate stressors are predominantly driven by fisheries stressors (targeted fishing and bycatch), particularly along coastlines and international waters just beyond the border of national Exclusive Economic Zones (Fig 2C).

Examining modeled impacts by quartile, rather than magnitude, allows for comparison of where particularly high impacts from climate stressors (which cannot be mitigated in the short run, but may benefit from adaptation) overlap with high impacts from non-climate stressors (which can be effectively mitigated through actions such as marine protected areas or fisheries management). Reclassifying impacts across these two aggregated categories into quartiles, based on global values, reveals 10.3% of ocean area where climate impact hot spots (i.e., spatial cells in the top quartile of global aggregated impact within the category) overlap with hot spots of non-climate impacts (e.g., Southeast Asia, East China Sea, Gulf of Mexico, Caribbean Sea, international tropical Pacific waters, Barents Sea, Bering Sea, Fig 2D), and 8.8% of area where cool spots (i.e., bottom quartile) of climate and non-climate impacts overlap (e.g., Southern portions of Pacific, Atlantic, and Indian oceans; Weddell Sea; national waters for several southern Pacific and Atlantic small island states, Fig 2D). These hot spots and cool spots represent the two most common instances of impact quartile overlap; conversely, high climate/low non-climate (2.9%) and low climate/high non-climate (3.1%) represent the rarest instances of overlap (S3A Fig in S1 File).

Patterns of predicted impacts based on the species method vs. representative habitats method (Fig 3; see S3B and S4 Figs in S1 File. for habitat analog to Fig 2 and S3A Fig in S1 File. respectively) show some clear differences, driven by variations in vulnerability across species and ecosystem type, despite the underlying stressor distributions being largely the same. Comparing overlap between habitat and species methods of climate impacts (Fig 3A) reveals areas where the two methods disagree (Sørensen similarity index 61%), indicating the species

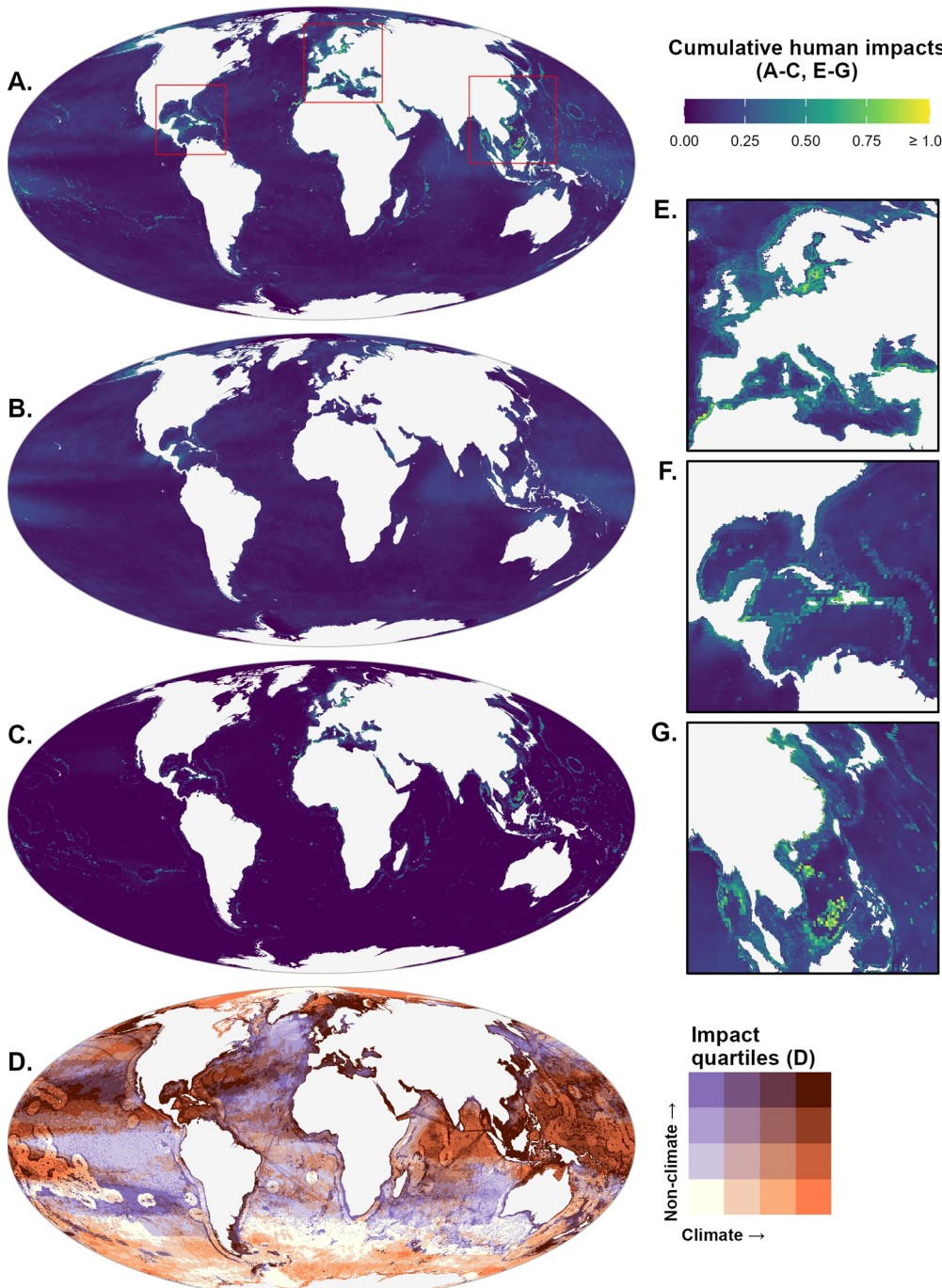

**Fig 2. Distribution of modeled risk of impact based on species-specific vulnerability and exposure to anthropogenic stressors.** (A) Mean cumulative impact across all species, summing across all stressors. (B) Mean cumulative impact across all species, summing across all climate-related stressors. (C) Mean cumulative impact across all species, summing across all non-climate stressors. (D) Bivariate comparison of distributions of climate impacts (orange) vs. non-climate impacts (purple) by quartile within each stressor group.

method predicts greater risk from climate change in equatorial Indian and Indo-Pacific waters (purple tones) relative to the habitat method, largely due to the inclusion of species-specific impacts as sea surface temperature rises relative to species thermal tolerance. Results based on

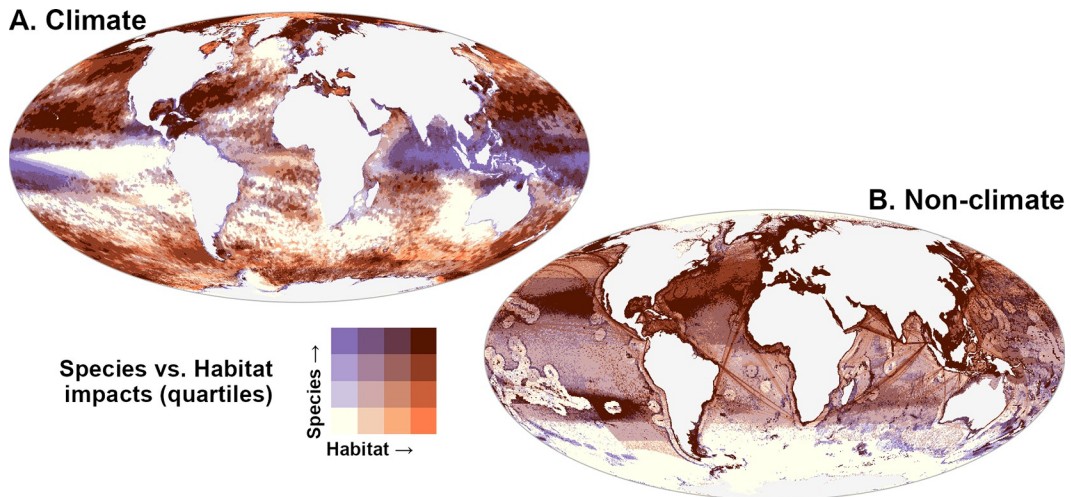

**Fig 3. Comparison of mean impacts from climate and non-climate stressors on marine ecosystems based on species and representative habitat methods.** Overlap of quartiles of impact from (A) climate stressors and (B) non-climate stressors, highlighting top quartile according to species-based cumulative impact (purple tones) and habitat-based cumulative impact (orange tones).

the habitat and species methods agree to a greater extent when examining non-climate stressors (Fig 3B, Sørensen similarity index 81%).

To facilitate comparison between the species and habitat methods, we converted impacts to percentiles and used these ranks to examine patterns within broad ecological realms [46]. Impacts varied considerably by method globally and by ecological realm, and often diverged strongly when comparing impacts on coastal areas (≤200 m depth) vs. open oceanic areas (Fig 4, by ecological province in S5 Fig in S1 File). As expected due to concentration of human activity along coastlines, we found that on average impacts for non-climate stressors (fishing, shipping, and land-based) in coastal areas dominated those in oceanic areas globally and across all realms, with general agreement in ranking between the habitat and species methods (Fig 4C). For climate stressors, the species method predicted higher relative impacts than the habitat method for coastal waters in all realms, particularly for Indo-Pacific regions, Tropical Atlantic and East Pacific, and the Southern Ocean (Fig 4B). The intersection of elevated climate impacts and non-climate impacts in species-rich coastal regions suggests a far higher risk to biodiversity than previously understood from habitat-based cumulative impact methods.

Comparing the species approach to the functional entity approach, we unsurprisingly see far greater agreement among quartiles for both climate stressors (Sørensen similarity index 75%) and non-climate stressors (Sørensen similarity index 93%) (Fig 5A and 5B), relative to the comparison between the species approach and habitat approach (Fig 3). As the functional entity method is a weighted mean of the same data underlying the (unweighted) species method, the two show substantial collinearity (climate stressors: $adj.R^2 = 0.959$, $p \ll .001$; non-climate stressors: $adj.R^2 = 0.923$, $p \ll 0.001$). Examining the difference between the two identifies locations where functionally vulnerable entities, i.e. FEs locally represented by a small number of species, are predicted to be at greater or lesser risk of impact than the overall average (Fig 5C and 5D). For much of the Atlantic, Indian, and Southern Oceans, on average, functionally vulnerable entities appear to face elevated risk of impact from climate stressors relative to the mean across all species, while the reverse appears true in the tropical western Pacific Ocean, the Caribbean Sea, and the Bay of Bengal. On average, functionally vulnerable entities appear to face lower impacts from non-climate stressors, particularly fishing. A notable

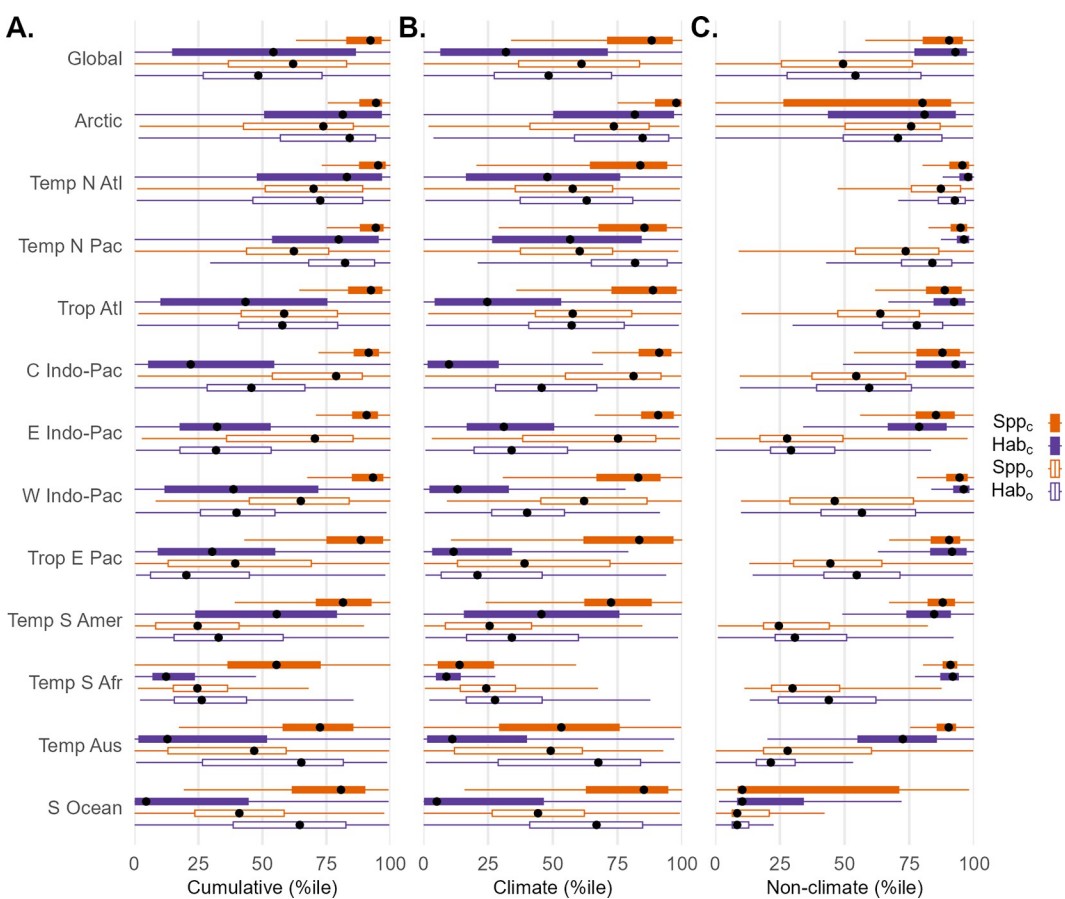

**Fig 4. Comparison of cumulative, climate, and non-climate stressors by habitat and species methods across 10 km resolution cells within coastal ($Spp_c$, $Hab_c$, solid boxes) and oceanic ($Spp_o$, $Hab_o$, hollow boxes) portions of 12 representative marine ecological realms, transformed to percentile ranks relative to global distribution within each impact category.** Black point indicates median value; boxes represent interquartile range (IQR, quartile Q1 to Q3); whiskers indicate observations 1.5x IQR below (above) Q1 (Q3) of box. Outliers omitted from plots for clarity. A) Cumulative impacts by species and habitat cumulative impact methods. B) Climate impacts by species and habitat methods. C) Non-climate impacts by species and habitat methods.

exception is apparent in the region between North America and Hawaii, due to high catch levels of Pacific skipjack tuna (*Katsuwonus pelamis*), which is the sole local representative of its functional entity in our dataset.

To determine the statistical significance of the difference between species and functional entity approaches, for each stressor, we calculated a 95% confidence interval around the difference by resampling species in each of 100,000 randomly selected cells and recalculating both the species and FE impact values 1000 times, retaining the 25th and 975th value (2.5% and 97.5%) as the 95% confidence interval. Samples where the 95% CI did not include zero were deemed statistically significant (p < 0.05). Fig 6 shows locations where one or more stressors indicated significantly higher (Fig 6A) or lower (Fig 6B) impacts based on the functional entity approach, relative to the species approach. At the cumulative impact level, fewer locations indicated statistical significance, at least in part because higher impacts from one or more stressors are often balanced by lower impacts from other stressors. Note that the elevated impacts from the skipjack tuna fishing seen in Fig 5D do not appear to be statistically significant according to this resampling method (Fig 6A), because such a difference driven by a single species is likely to be highly sensitive to resampling.

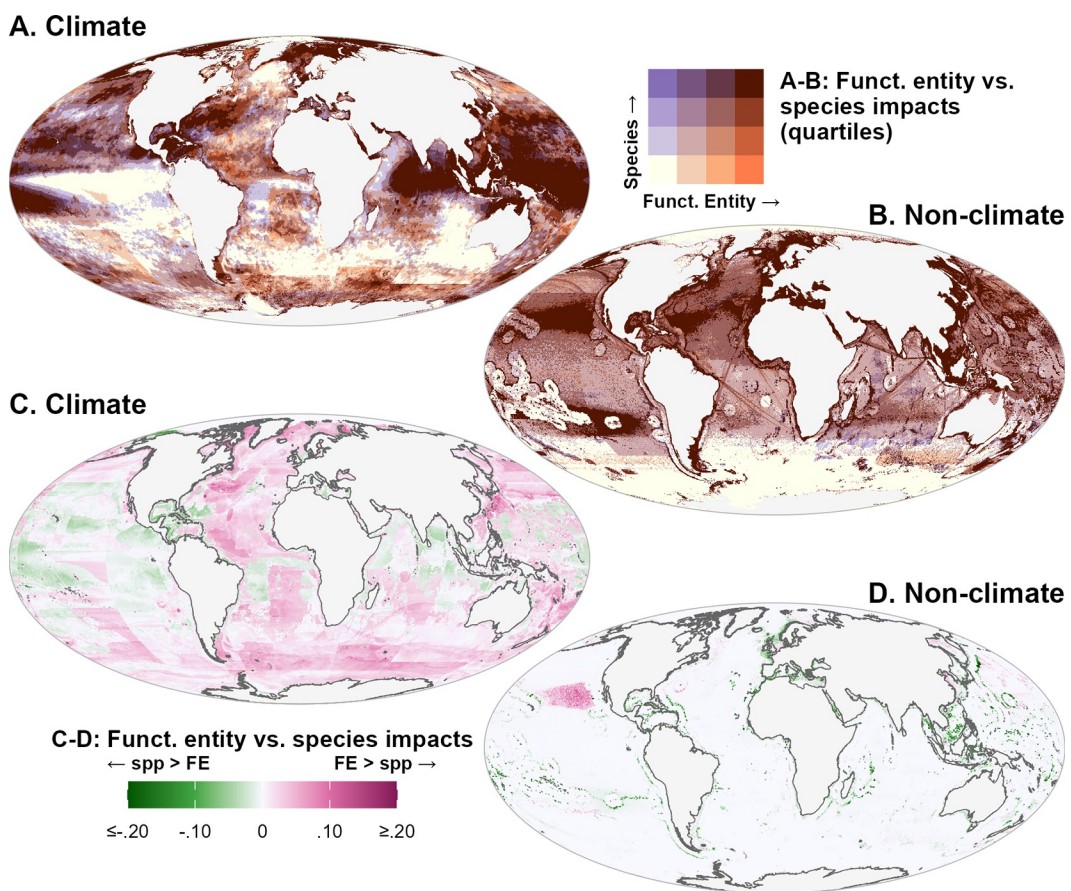

**Fig 5. Comparison of mean impacts from climate and non-climate stressors on marine ecosystems based on species and functional entity methods.** Panels A and B show overlap of quartiles of impact from (A) climate stressors and (B) non-climate stressors, highlighting top quartile according to species-based cumulative impact (purple tones) and habitat-based cumulative impact (orange tones). Panels C and D show the difference in mean impacts between the functional entity approach and species approach for (C) climate stressors and (D) non-climate stressors; magenta tones indicate areas where impacts on vulnerable functional entities are higher than average across all species, while green tones indicate lower impacts on vulnerable functional entities than average across all species.

## Discussion

The CBD Kunming-Montreal Global Biodiversity Framework (GBF) proposes that at least 30% of ocean areas are under some form of effective marine protection by 2030, with an emphasis on areas critical for biodiversity and its contributions to people [12]. Global and regional studies of marine biodiversity have presented different aggregation methods to communicate conservation-relevant information, including population density, endemicity, extinction risk, taxonomic group, or simultaneous ecological, evolutionary, and social domains (e.g., [3, 47–51]). In all cases, the purpose of aggregation is to effectively synthesize and communicate a complex and multifaceted set of variables, and each aggregation method bears its own advantages and disadvantages. Our equal-weighted approach to species cumulative impact estimation is simple to calculate and understand, and accounts for the fact that the extinction of any species is likely detrimental to a functional ecosystem. Our functional entity approach integrates information about functional redundancy and vulnerability to prioritize protection of ecosystem function and delivery of ecosystem services. These species-based approaches to estimating cumulative impacts, in conjunction with the well established habitat-

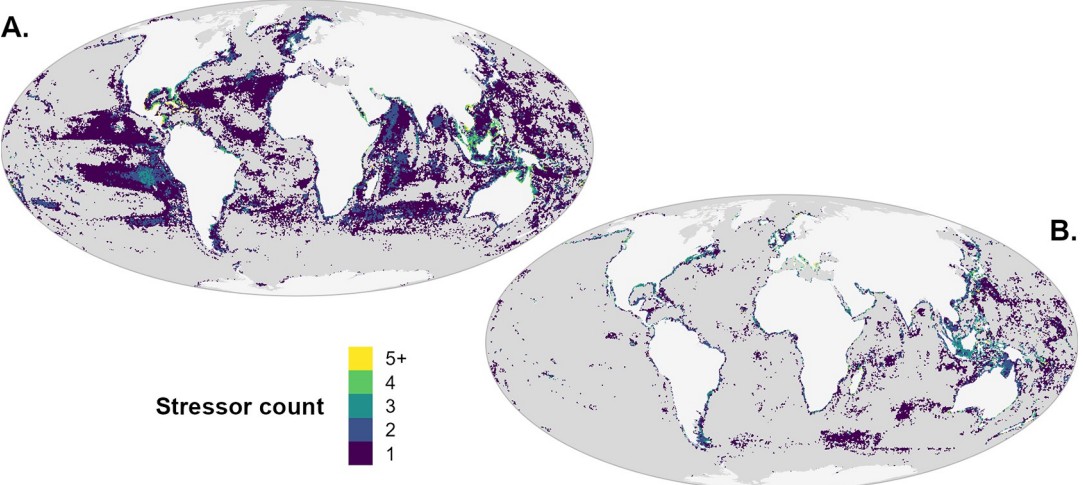

**Fig 6. Statistically significant areas of difference in impact between the functional entity approach and species approach.**
(A) Number of stressors per pixel in which the functional entity approach indicated significantly higher (p < 0.05) impacts than the species approach. (B) Number of stressors per pixel in which the functional entity approach indicated significantly lower (p < 0.05) impacts than the species approach. Grey pixels indicate areas where no stressors resulted in significantly higher/lower impacts according to the functional entity approach.

based approach [7, 14], provide a more complete understanding of vulnerability, risk, and impact to marine ecosystems, providing valuable information that can help inform progress toward the GBF's protection targets.

Our results show that impacts from climate-related stressors in general dominate impacts from non-climate stressors, regardless of the ecosystem element studied. This is due to broad exposure to elevated sea surface temperature and ocean acidification coupled with widespread vulnerability of species to these stressors. Even under the most optimistic emissions reduction scenarios, climate stressors are expected to increase in intensity in the near term [52–54]. While local conservation policy cannot mitigate climate stressors driven by global emissions, it can reduce the intensity of stressors directly related to human activity, especially fishing, shipping, nutrient runoff, and coastal development, giving impacted species a greater chance at surviving, recovering from, or adapting to the effects of rising temperatures and ocean acidification [55, 56]. Overlapping hotspots of high climate impact between the species and habitat approaches (Figs 3 and 4) indicate areas where high intensity stressors intersect with vulnerable species and habitats, indicating ecosystems at greatest risk of biodiversity declines. Areas of overlap of high-intensity impacts from both climate and non-climate stressors should be prioritized for action to curtail human activities to reduce the risk of ecosystem collapse.

Because the stressor layers used to estimate impacts by each method are largely identical, substantial differences in predicted impact among methods generally result from differences in underlying vulnerability of the ecosystem elements of interest: species, habitat, or functional entity. The species-based approach enables us to attribute risk of impact to particular species-stressor interactions, rather than ecosystem-stressor interactions as for the habitat-based approach. In regions where the habitat approach predicts higher impacts than the species or functional entity approach, the species and functions within the ecosystem may be individually quite robust to the stressors present, but the broader processes and interactions that govern ecosystem health, or taxa not included in the species-based approach, may be adversely affected. In such cases, such as Hawai'i's oceanic waters in the Eastern Indo-Pacific realm, blanket protections such as the fully protected Papahānaumokuākea Marine National

Monument can limit human activity to protect ecosystem services [55, 57] and increase resilience to system-wide climate impacts [56].

Conversely, where a species or functional entity approach predicts higher impacts than the habitat approach, the general structure of the ecosystem may be robust to the stressors present, but the region may be home to one or more highly impacted species, increasing extinction risk and posing greater risk of loss of ecosystem function. Such a pattern is evident in coastal waters across the globe, driven largely by climate impacts (Fig 4), suggesting greater risk to coastal biodiversity than previously understood from habitat-based methods. In these cases, targeted management to reduce impacts on that subset of species may be ecologically effective while remaining more politically and economically attractive than full exclusion, including both ocean-based and land-based interventions [58, 59]. For example, the Andaman Islands in the West Indo-Pacific realm (Fig 4 Western Indo-Pacific, SI) did not experience substantial SST extreme events during the years of our data, but annual mean SST has risen such that many species in vulnerable functional entities are near the top of their thermal tolerances, elevating risk of extirpation. This includes several species of mullet harvested in small-scale gillnet fisheries, as well as many benthic molluscs and polychaetes subject to bottom trawl impacts, so targeted gear restrictions could potentially reduce fishing pressure on these climate-stressed species without requiring full closure.

Our approach to predicting human impacts systematically across many diverse species provides valuable insight to help reduce extinction rates and risk of marine species, a key goal of the GBF [12]. Two other key goals would be well served by considering species in context of their ecosystem roles and functions: securing the integrity and resilience of ecosystems, and ensuring the sustainability of nature's contributions to people through ecosystem functions and services [12]. Our functional entity-weighted approach to predicting impacts strives to provide insights to support these goals by integrating information on functional redundancy and vulnerability. However, this approach requires far more information than the equal-weighted species approach and requires more complex assumptions. In particular, the assignment of species to functional entities may be quite sensitive to the choice of traits and categorical bins for trait values that distinguish one functional entity from another. A sensitivity analysis based on random resampling of functional trait values showed that in aggregate, the cumulative impact results changed little, and in most areas actually increased slightly as random shuffling generated functional trait combinations unlikely to occur in nature, resulting in virtual functional entities represented by only one or two species (SI Methods in S1 File, S6 Fig in S1 File).

Comparing the species and functional entity approaches shows that for much of the ocean, weighting impacts by functional vulnerability produces similar patterns to the equal-weighted species approach in rank (Fig 5A and 5B), generally small differences in magnitude (Fig 5C and 5D), and strong linear correlations. This suggests that the species-mean approach captures similar patterns of impact with far greater parsimony and thus can be a useful first-pass proxy for impacts to functional diversity. However, statistically significant differences in mean impact (though potentially small in magnitude) at the stressor level were seen in 39% of the ocean (Fig 6). To understand the risk of impact to ecosystem functions and services, we must sharpen our understanding of functional diversity and how it interacts with anthropogenic stressors. Whether applying a categorical functional entity approach or continuous functional diversity metrics based on distribution in multidimensional trait space (e.g., [28, 60, 61]), such an endeavor requires continuing effort to collect and make available data on ecologically relevant traits [62].

For several reasons, our results may be conservative. First, the anthropogenic stressors included in this analysis are by no means the only ways in which humans impose adverse

effects on marine ecosystems [6, 15]; however, while vulnerability estimates may be available for a broader suite of stressors [15, 19], our analysis was necessarily limited to stressors whose human-driven deviations from natural levels have been mapped globally. Important and/or emerging stressors such as marine plastics, noise pollution, persistent organic chemical pollutants, human-driven sedimentation, and seabed mining [6] should be priorities for future cumulative human impact models, especially as there may be localized contexts in which the impact of such stressors may outweigh those included in our study. Second, we assume that impact scales linearly with stressor intensity, yet it is likely that individual, population, and community-level responses to stressors include thresholds, nonlinearities, and dynamic ecosystem interactions leading to accelerating marginal risk [63, 64]. Finally, interactions among multiple simultaneous stressors may result in synergistic impacts rather than the simple additive model we have incorporated here [34, 35, 63, 65], though such synergies remain an area of high uncertainty [6].

While effective and equitable conservation efforts must be well grounded in local and regional knowledge and values [66], global scale assessments such as this are necessary to inform the global biodiversity conservation agenda, provide broader context for local decision-making, and understand ecological and political synergies and tradeoffs across scales [67]. While our results are useful for conservation and decision-making, equally important is the framework used to generate those results. The methods presented here can be adapted to local and regional scales, incorporating finer-resolution data on species ranges, stressors, and species traits where available, to better inform local conservation decisions to meet the needs, values, and priorities of local stakeholders and rights-holders.

Looking forward, rising sea surface temperatures in particular are predicted to impose substantial impacts on species as mean temperatures on warm/equator-ward range limits rise and exceed species thermal preferences, especially for tropical species as many have evolved narrow thermal ranges due to relatively stable year-round temperatures [68]. Climate-driven shifts in species ranges are likely to shift patterns of vulnerability and impact over the next decades, potentially driving vulnerable but currently unexposed species into the path of higher-intensity stressors or opening up new habitat that provides refuge for highly impacted species from current stressors. Understanding future patterns of vulnerability in conjunction with expected changes in anthropogenic stressors must be a key concern for designing effective and lasting conservation strategies [6]. The present analysis does not account for expected climate-driven range shifts, though the impacts predicted from SST rise reflect a major mechanism driving poleward retreats of warm trailing range edges. Future research could incorporate projections of climate-driven species range shifts (e.g., [5, 69, 70]) with forward-looking models or simulations to account for uncertainty, reference conditions, and dynamic changes in disturbance regimes [24] to predict impacts on novel range as species cold leading range edges expand into ever more temperate poleward waters.

Conservation is ultimately about balancing the social, cultural, and economic benefits and costs of conservation to improve or maximize overall utility for humans, including sustainable provision of natural resources or gainful employment, long-term delivery of ecosystem services at the local or global scale, and protecting nature to ensure its continued existence for future generations to enjoy [10]. In light of the Kunming-Montreal Global Biodiversity Framework [12], as we strive to protect 30% or more of our ocean by 2030, we must apply a holistic approach to conservation to prevent the loss of critical ecosystems, protect the functional diversity that underpins resilience and ecosystem services, slow or halt species declines and extinctions, and maintain genetic diversity essential for long-term adaptation [9]. This is especially important as climate impacts are already disrupting ecosystems and will continue to increase for decades, even under the most ambitious emissions reduction scenarios [52, 53],

necessitating conservation action to mitigate non-climate stressors (e.g., reduction of land-based runoff) to allow for improved ecosystem resilience to climate change [55, 56]. While well-enforced no-take marine protected areas are an effective conservation tool that can provide multiple co-benefits [55, 71–73], in certain cases, sustainable-use marine protected areas with targeted exclusions may provide substantial ecological benefit at lower social cost [74]. Consideration of human impacts across lenses of species, function, and habitat provides a richer understanding of marine ecosystems, and highlights that impacts in species-rich coastal regions may pose greater risk to biodiversity than indicated from habitat-based methods alone. Our data can be used with socioeconomic information to help prioritize effective, economically efficient, and socially equitable conservation actions to best benefit nature and people.

## Supporting information

**S1 File.**
(PDF)

## Acknowledgments

We thank the National Center for Ecological Analysis and Synthesis (NCEAS) for computational support.

## Author Contributions

**Conceptualization:** Casey C. O'Hara, Melanie Frazier, Nathalie Butt, Carissa Klein, Benjamin S. Halpern.

**Data curation:** Casey C. O'Hara.

**Formal analysis:** Casey C. O'Hara.

**Funding acquisition:** Carissa Klein, Benjamin S. Halpern.

**Methodology:** Casey C. O'Hara, Melanie Frazier, Mireia Valle, Nathalie Butt, Carissa Klein, Benjamin S. Halpern.

**Software:** Casey C. O'Hara.

**Supervision:** Carissa Klein, Benjamin S. Halpern.

**Visualization:** Casey C. O'Hara.

**Writing – original draft:** Casey C. O'Hara.

**Writing – review & editing:** Casey C. O'Hara, Melanie Frazier, Mireia Valle, Nathalie Butt, Kristin Kaschner, Carissa Klein, Benjamin S. Halpern.

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
