## [Decision Letter · Decision Letter 0]

17 Jul 2024

PONE-D-24-24618Cumulative human impacts on global marine fauna highlight risk to biological and functional diversityPLOS ONE

Dear Dr. O'Hara,

Thank you for submitting your manuscript to PLOS ONE. After careful consideration, we feel that it has merit but does not fully meet PLOS ONE’s publication criteria as it currently stands. Therefore, we invite you to submit a revised version of the manuscript that addresses the points raised during the review process.

We look forward to receiving your revised manuscript.

Kind regards,

Abdul Azeez Pokkathappada, Ph.D.

Academic Editor

PLOS ONE

Journal Requirements:

3. We note that Figures 2,3,5 and 6 in your submission contain map/satellite images which may be copyrighted. All PLOS content is published under the Creative Commons Attribution License (CC BY 4.0), which means that the manuscript, images, and Supporting Information files will be freely available online, and any third party is permitted to access, download, copy, distribute, and use these materials in any way, even commercially, with proper attribution. For these reasons, we cannot publish previously copyrighted maps or satellite images created using proprietary data, such as Google software (Google Maps, Street View, and Earth). For more information, see our copyright guidelines: http://journals.plos.org/plosone/s/licenses-and-copyright.

a. You may seek permission from the original copyright holder of Figures 2,3,5 and 6  to publish the content specifically under the CC BY 4.0 license.  

Reviewers' comments:

Reviewer's Responses to Questions

**Comments to the Author**

1. Is the manuscript technically sound, and do the data support the conclusions?

Reviewer #1: Yes

Reviewer #2: Yes

2. Has the statistical analysis been performed appropriately and rigorously? 

Reviewer #1: Yes

Reviewer #2: Yes

3. Have the authors made all data underlying the findings in their manuscript fully available?

Reviewer #1: Yes

Reviewer #2: Yes

4. Is the manuscript presented in an intelligible fashion and written in standard English?

Reviewer #1: Yes

Reviewer #2: Yes

5. Review Comments to the Author

Reviewer #1: The authors assess threats to marine life by developing, presenting and comparing metrics of human impact as seen through the lens of species, habitat and functional diversity. Bringing together these different analyses at the global level and in a way that allows them to be inter-compared is a new and useful addition to the literature. The paper is well written, interesting and potentially important – I enjoyed reading it and I’m happy to recommend it for publication with a few minor corrections.

The introduction is concise and informative, placing the study clearly in the context of previous literature.

The methods section largely sets out the data sources and calculations in sufficient detail, but it’s not clear how the stressor intensities are assigned. It would be helpful to include a few sentences outlining the data sources and a reference to the supporting information. In particular, are the climate stressors defined in relation to present-day conditions or some projected future?

Line 149: I’m not sure why organic chemical pollutants is the only stressor not included in the study which is mentioned at this point; others are (rightly) mentioned in the discussion, line 470-471.

The various impact scores are created by adding the effects of single stressors and so they cannot include compound effects of more than one stressor acting in combination. This is raised in the discussion, but I suggest that it would be helpful to also acknowledge it in the methods section, with a brief explanation of your reasons for this choice.

The results section presents a selection of well-designed figures which nicely bring out the main points of this complicated study. I just have a few small points:

Figure 2: the spatial differences don’t come out vary clearly in A and B – perhaps the color scale could be adjusted to give more contrast to mid-range values.

Line 284: I’m not sure why quantiles are mentioned, when all the results are presented in terms of quartiles.

Lines 327-329: the point about the species method suggesting a higher risk to biodiversity than previous habitat-based analyses seems important, and worth moving to the discussion or mentioning again there.

The discussion is consistent with the results and includes all the points I was hoping would be raised - I have no suggestions for improvement.

Supporting information: The equations have not been rendered correctly. This may just be an issue in the review copy, but worth checking.

Reviewer #2: ****Summary****

The authors present a well-considered study that provides a global overview of species, habitat, and functional entity vulnerability to climate and non-climate anthropogenic stressors. The approach is fairly straightforward, but the methods are missing several key details necessary to understand the nuances associated with the results.

****General Comments/Questions****

To understand how the impact of a stressor on a species was calculated (Ii in equation 1) required going to the Supporting Information, then to Butt et al. (2022), then to the Supporting Information in Butt et al. (2022). Is there a way to streamline this pathway?

The assessment builds from the impact of individual stressors on individual species, which is derived from vulnerability of the species to that stressor. It is unclear which traits were used for the sensitivity and adaptive capacity terms within vulnerability in the present study. Considering these values represent the foundation for the rest of the analysis, is there any estimate of the uncertainty of those trait values? How are ranges of values handled, such as intraspecific variation or regional differences?

****Line comments****

Line 103. Is the full list of species included in the analysis available anywhere? It is useful to know which species met the necessary criteria for inclusion.

Lines 113-122. How sensitive is the assessment to the 0.5 probability of occurrence value applied to the AquaMaps? In cases where both IUCN and AquaMaps data were available, how closely did the calculated AquaMaps overlap with the IUCN? Essentially, is it reasonable to consider these two sources similar enough to aggregate them?

Line 124. Were all species included in the current assessment also assessed in Butt et al. (2022)?

Line 133. SI Methods – Vulnerability Estimates. Gi/Gi,max is an undefined term with no associated text elsewhere in the SI. In the original methodology (Butt et al. 2022), Gi represented general adaptive capacity summed over a series of traits. After the equation, the authors present two modifications to the Butt et al. 2022 methodology. They mention body length, fecundity, generation time, temperature tolerances, and depth preferences in addition to extent of occurrence. It is unclear if these traits were used only for those species missing traits in the original Butt et al. (2022) study or if only those traits were used for all species in the present study. Additionally, it is unclear if adaptive capacity values were sourced from Butt et al. (2022) or if they were determined anew in the present study.

Line 133. SI Methods – Vulnerability Estimates. How did the vulnerability scores in this study compare with the vulnerability scores from Butt et al. (2022)? Did the reduced number of traits affect vulnerability scores in a meaningful way?

Lines 138-148. Please describe how stressors were determined to be universal or “per species.” UV radiation, sea level rise, light pollution, and ship strike would also seem dependent on position in the water column in a similar manner as bycatch.

Lines 163-164. How does species-specific thermal stress not apply at the habitat level for things like coral reefs, kelp forests, and seagrass beds?

Lines 168-170. Here, but possibly earlier with the vulnerability estimates, it would be worth mentioning that the exposure term in the vulnerability estimate is a 0 or 1 potential indicator and does not relate to the level or magnitude of exposure. Otherwise, stressor intensity (s sub j) appears to be double counting exposure to a stressor.

Lines 196-197. How was ph (p sub h) determined?

Lines 276-280. Can you also highlight areas with the least impact?

Lines 288-295. Possibly in the SI, it would be helpful to see the percent occurrence in each cell of this quartile x quartile matrix presented in Fig 2D in addition to highlighting the cell values of the two most extreme ends of the matrix in the text.

Lines 326-329. This seems like a major conclusion from the paper. I would expect to see this mentioned in the abstract and in the concluding paragraphs.

Lines 411-413. This is a methodological decision that could probably be better explained/justified. Given that these estimates may be synergistic, why use an additive approach instead of one that could include those potential synergies?

Lines 465-471. Are there any examples where locally important stressors that were not included due to data coverage may have outweighed the impact measured by those stressors included in the current study?

Lines 474-477. This is repetitive of lines 411-413. Including with this paragraph does highlight that this decision may result in more conservative results, but additional rationale for the decisions would provide necessary context.

Lines 516-520. I can readily see how this approach can inform decision-makers about locations for conservation but it’s less clear how this can inform strategies for conservation. Can the authors provide examples of conservation strategies beyond MPAs that this work would inform given spatial resolution, species included, and climate/non-climate stressors considered? Within these 4x4 matrices presented in figures 2D, 3A/B, and 5A/B, is there a prioritization scheme for MPAs or other area-specific management strategies?

6. PLOS authors have the option to publish the peer review history of their article (what does this mean?). If published, this will include your full peer review and any attached files.

Reviewer #1: No

Reviewer #2: No

---

## [Author Response · Author response to Decision Letter 0]

2 Aug 2024

Casey O’Hara

National Center for Ecological Analysis and Synthesis

1021 Anacapa St.

Santa Barbara CA 93101

ohara@nceas.ucsb.edu

August 1, 2024

Dear Dr. Pokkathappada,

My coauthors and I thank you and your reviewers for the thorough and thoughtful review of our manuscript, “Cumulative human impacts on global marine fauna highlight risk to biological and functional diversity.” We have addressed all the journal requirements and reviewers’ comments, and feel that the resulting manuscript is now stronger as a result. Below, we describe the changes we have made (in serif font face) in response to the reviewers’ comments (in italic sans serif face).

We believe our submission meets all the PLOS ONE style requirements. If there are specific items that we have not met, please provide more detailed information so we can properly address them.

We have uploaded the Github repository containing all code and output data (as a single .zip file) to Figshare, along with a folder containing the primary output rasters as .tif files. This collection has been assigned a DOI (10.6084/m9.figshare.26454106) and we will publish this collection upon acceptance (in case any additional review necessitates further changes).

3. We note that Figures 2,3,5 and 6 in your submission contain map/satellite images which may be copyrighted.

All maps figures are original to our analysis, using the code and data generated for our study, and do not contain any copyrighted information. All maps included in figures use land form polygons from the Natural Earth collection (https://www.naturalearthdata.com/), which is public domain and does not require citation (though we have cited the Natural Earth website in the text of the Supporting Methods). 

We have provided our Supporting Information as a single PDF containing supporting figures, tables, and methods. We believe our format adheres to the guidelines, and have added a caption for the file at the end of the manuscript (after acknowledgments, before references):

“S1 Supporting Information. Supporting figures, tables, methods, and references. Supporting information includes: Figures S1 – S7, Tables S1 – S4, Supporting Methods, and Supporting References (75-98).”

We believe our reference list is complete and correct, and are not aware of any cited papers that have been retracted. Several new citations have been added to address the reviewers’ comments, and these have been noted in the responses to those comments below.

Reviewer #1

The authors assess threats to marine life by developing, presenting and comparing metrics of human impact as seen through the lens of species, habitat and functional diversity. Bringing together these different analyses at the global level and in a way that allows them to be inter-compared is a new and useful addition to the literature. The paper is well written, interesting and potentially important – I enjoyed reading it and I’m happy to recommend it for publication with a few minor corrections.

We thank the reviewer for the kind words. We appreciate their constructive feedback and hope we have sufficiently addressed their questions and suggestions.

The introduction is concise and informative, placing the study clearly in the context of previous literature. The methods section largely sets out the data sources and calculations in sufficient detail, but it’s not clear how the stressor intensities are assigned. It would be helpful to include a few sentences outlining the data sources and a reference to the supporting information. In particular, are the climate stressors defined in relation to present-day conditions or some projected future?

We have added additional information to the methods in the main text describing how stressor intensities are assigned (lines 142-149):

“Spatial data for stressors relates to some physical quantity related to anthropogenic activity, e.g., brightness of nighttime lights, tonnes of nutrient fertilizer runoff, population density within 25 km of coast, or value of aragonite saturation state. For each stressor, a reference value was determined from the data (typically 99.9th percentile of observed values), a historic baseline (e.g., mean/standard deviation of sea surface temperature from 1985-2015), or ecologically relevant value (e.g., aragonite saturation state of 1) (Table S4). We calculated stressor distributions as a value from 0 (stressor not present) to 1 (stressor at reference point, indicating maximum intensity).”

Line 149: I’m not sure why organic chemical pollutants is the only stressor not included in the study which is mentioned at this point; others are (rightly) mentioned in the discussion, line 470-471.

We thank the reviewer for calling this out. Our present analysis builds on a similar (habitat-focused) analysis in Halpern et al. (2019), which did include organic chemical pollutants. This mention in the methods was originally intended to highlight this difference from the 2019 study. However, it does seem out of place in the methods, so we have removed this specific mention, while retaining the text in the discussion explaining the reason for omitting this stressor.

The various impact scores are created by adding the effects of single stressors and so they cannot include compound effects of more than one stressor acting in combination. This is raised in the discussion, but I suggest that it would be helpful to also acknowledge it in the methods section, with a brief explanation of your reasons for this choice.

The reviewer’s suggestion was also echoed by Reviewer 2. We have added text to the methods in the section describing the Cumulative Human Impacts: species method (lines 186-190):

“Note that this additive model does not account for compound effects of multiple stressors acting in combination, i.e., synergistic or antagonistic effects. Meta-analyses examining two-stressor interactions (Crain et al. 2008; Stockbridge et al. 2020) have observed some non-additive stressor interactions, but additive effects were more commonly reported. Additionally, an additive model requires fewer assumptions, is conceptually tractable, and likely results in more conservative results.”

The results section presents a selection of well-designed figures which nicely bring out the main points of this complicated study. I just have a few small points:

Figure 2: the spatial differences don’t come out vary clearly in A and B – perhaps the color scale could be adjusted to give more contrast to mid-range values.

We had used a non-linear scale on 2A, B, and C to help visualize differences in the low- to mid-range impact areas, but this deemphasizes the areas where they are particularly high, reducing that contrast. We have changed the scale back to a linear scale but capped the values at the 99.99th percentile (to reduce the influence of outliers on the color scale) and added inset maps showing regions of interest and high variation to the figure. We have updated the analogous habitat map in the supporting figures as well.

Line 284: I’m not sure why quantiles are mentioned, when all the results are presented in terms of quartiles.

We thank the reviewer for pointing this out this inconsistency. We’ve changed this instance to “quartile.”

Lines 327-329: the point about the species method suggesting a higher risk to biodiversity than previous habitat-based analyses seems important, and worth moving to the discussion or mentioning again there.

This point was noted by both reviewers, and we have added text to the abstract, discussion, and concluding paragraph to help highlight this result:

Abstract (39-42): “Comparing species-level modeled impacts to those based on marine habitats that represent important marine ecosystems, we find that even relatively untouched habitats may still be home to species at elevated risk, and that many species-rich coastal regions may be at greater risk than indicated from habitat-based methods alone.”

Discussion (456-461): “Conversely, where a species or functional entity approach predicts higher impacts than the habitat approach, the general structure of the ecosystem may be robust to the stressors present, but the region may be home to one or more highly impacted species, increasing extinction risk and posing greater risk of loss of ecosystem function. Such a pattern is evident in coastal waters across the globe, driven largely by climate impacts (Fig. 4), suggesting greater risk to coastal biodiversity than previously understood from habitat-based methods.”

Conclusion (551-555): “Consideration of human impacts across lenses of species, function, and habitat provides a richer understanding of marine ecosystems, and highlights that impacts in species-rich coastal regions may pose greater risk to biodiversity than indicated from habitat-based methods alone.”

The discussion is consistent with the results and includes all the points I was hoping would be raised - I have no suggestions for improvement.

We appreciate the reviewer’s feedback and we are glad that our discussion has met their expectations.

Supporting information: The equations have not been rendered correctly. This may just be an issue in the review copy, but worth checking.

Looking at the Microsoft Word .docx version that we submitted, with one exception (due to a typo which has been fixed), the equations all appear to be rendered correctly; perhaps as the reviewer suggested this may just be a problem with the review copy. We will verify that the equations are rendered correctly when we resubmit the revised version of the manuscript and supporting information.

Reviewer #2

Summary

The authors present a well-considered study that provides a global overview of species, habitat, and functional entity vulnerability to climate and non-climate anthropogenic stressors. The approach is fairly straightforward, but the methods are missing several key details necessary to understand the nuances associated with the results.

We appreciate the reviewer’s feedback, and we feel our modifications to address their concerns have improved the manuscript, particularly the methods.

General Comments/Questions

To understand how the impact of a stressor on a species was calculated (Ii in equation 1) required going to the Supporting Information, then to Butt et al. (2022), then to the Supporting Information in Butt et al. (2022). Is there a way to streamline this pathway?

We thank the reader for this feedback. To help communicate the methods of intersecting species ranges, stressor intensities, and species-stressor vulnerabilities, we have included an additional figure in the Supporting Information to help conceptually explain the process. In addition, we have added a brief note to provide the reader an overview of the process before proceeding into the technical details (lines 85-88):

“For each species/stressor combination, we intersect the species' range with the spatial distribution of the stressor; impact in each pixel is modeled as the product of stressor intensity and the species' estimated vulnerability to that stressor. (See Fig. S1 for conceptual overview of methods)”

We feel the methods are far clearer thanks to these modifications to address the reviewer’s concern.

The assessment builds from the impact of individual stressors on individual species, which is derived from vulnerability of the species to that stressor. It is unclear which traits were used for the sensitivity and adaptive capacity terms within vulnerability in the present study. Considering these values represent the foundation for the rest of the analysis, is there any estimate of the uncertainty of those trait values? How are ranges of values handled, such as intraspecific variation or regional differences?

The full methodology of the trait-based vulnerability framework is detailed in Butt et al. (2022), including detailed answers to the questions noted by the reviewer. However, as the reviewer noted, this is fundamental to the rest of the analysis, and so we have included in our supporting information a table that links traits to the sensitivity and adaptive capacity to each of the stressors included in our study for easier reference.

The original trait-based framework (Butt et al. 2022) incorporated a sensitivity analysis around the numeric scores assigned to different trait values and found that varying the scoring scale (linear/convex/concave) generally resulted in changes in magnitude of vulnerability, but not rank (i.e., lower vulnerability species stayed low, higher stayed high). This does not capture uncertainty in trait values themselves, but it does reflect uncertainty in the translation of trait values to vulnerability scores, which is relevant to our purposes.

Finally, trait values for the trait-based framework (Butt et al. 2022) were elicited from experts and generally provided as a single value for a given species for a given trait. Many traits were categorical and broadly applicable across a taxon (e.g., respiration mechanisms such as gills vs. lungs) and thus intraspecific or regional variation is unlikely. For traits on a continuous scale (e.g., fecundity, body length, extent of occurrence), values were binned into ordinal categories which were generally quite broad, likely able to accommodate intraspecific variation and regional variation.

To address these concerns of the reviewer, we have added caveats and a fuller explanation of the trait methods in the supporting information:

“The expert-elicited trait values used to calculate vulnerability were given as ordinal or nominal categorical values, generally with a single value attributed to each trait for each species. While a single value does not allow for large-scale intraspecific variability or regional variations in traits, most traits are broadly applicable across a species, genus, or family (e.g., respiration structures of gills vs. lungs vs. diffusion), and for traits representing continuous values (e.g., body size, fecundity, age to maturity), the bins are typically in orders of magnitude, wide enough to accommodate substantial variation without affecting the vulnerability estimate.”

Line comments

Line 

---

## [Decision Letter · Decision Letter 1]

20 Aug 2024

Cumulative human impacts on global marine fauna highlight risk to biological and functional diversity

PONE-D-24-24618R1

Dear Dr. O'Hara,

We’re pleased to inform you that your manuscript has been judged scientifically suitable for publication and will be formally accepted for publication once it meets all outstanding technical requirements.

Kind regards,

Abdul Azeez Pokkathappada, Ph.D.

Academic Editor

PLOS ONE

Additional Editor Comments (optional):

Dear authors, please cross check the Table and Figure numbers including supplimentary material. Thank you.

Reviewers' comments:

Reviewer's Responses to Questions

**Comments to the Author**

1. If the authors have adequately addressed your comments raised in a previous round of review and you feel that this manuscript is now acceptable for publication, you may indicate that here to bypass the “Comments to the Author” section, enter your conflict of interest statement in the “Confidential to Editor” section, and submit your "Accept" recommendation.

Reviewer #1: (No Response)

Reviewer #2: All comments have been addressed

2. Is the manuscript technically sound, and do the data support the conclusions?

Reviewer #1: Yes

Reviewer #2: Yes

3. Has the statistical analysis been performed appropriately and rigorously? 

Reviewer #1: Yes

Reviewer #2: Yes

4. Have the authors made all data underlying the findings in their manuscript fully available?

Reviewer #1: Yes

Reviewer #2: Yes

5. Is the manuscript presented in an intelligible fashion and written in standard English?

Reviewer #1: Yes

Reviewer #2: Yes

6. Review Comments to the Author

Reviewer #1: Thanks to the authors for addressing my comments thoroughly. I would just like to highlight one small error - in the Supporting Information the third table is labelled Table S4.

Reviewer #2: The authors did great work responding to and addressing the comments. The additional figures and tables included in the supplementary material make the methods much more accessible to the reader and the associated text added to the main body greatly improve the clarity of the work. I have no additional concerns and look forward to seeing this work published.)

7. PLOS authors have the option to publish the peer review history of their article (what does this mean?). If published, this will include your full peer review and any attached files.

Reviewer #1: No

Reviewer #2: No

---

## [Editor Report · Acceptance letter]

22 Aug 2024

PONE-D-24-24618R1 

PLOS ONE

Dear Dr. O'Hara, 

I'm pleased to inform you that your manuscript has been deemed suitable for publication in PLOS ONE. Congratulations! Your manuscript is now being handed over to our production team.

Kind regards, 

on behalf of

Dr. Abdul Azeez Pokkathappada 

Academic Editor

PLOS ONE